# Challenges and strategies in the soluble expression of CTA1-(S14P5)4-DD and CTA1-(S21P2)4-DD fusion proteins as candidates for COVID-19 intranasal vaccines

Simson Tarigan[1]*, Gita Sekarmila[1], Apas[1], Sumarningsih[1], Ronald Tarigan[2], Riyandini Putri[1], Damai Ria Setyawati[1]

1 Research Organization for Health, National Research and Innovation (BRIN), Cibinong, Indonesia,
2 School of Veterinary and Medical Sciences, IPB University, Bogor, Indonesia

* sitariganta@gmail.com

**Data Availability Statement:** All relevant data are within the manuscript and its Supporting Information files.

## Abstract

Developing intranasal vaccines against pandemics and devastating airborne infectious diseases is imperative. The superiority of intranasal vaccines over injectable systemic vaccines is evident, but developing effective intranasal vaccines presents significant challenges. Fusing a protein antigen with the catalytic domain of cholera toxin (CTA1) and the two-domain D of staphylococcal protein A (DD) has significant potential for intranasal vaccines. In this study, we constructed two fusion proteins containing CTA1, tandem repeat linear epitopes of the SARS-CoV-2 spike protein (S14P5 or S21P2), and DD. Structural predictions indicated that each component of the fusion proteins was compatible with its origin. *In silico* analyses predicted high solubility for both fusion proteins when overexpressed in *Escherichia coli*. However, contrary to these predictions, the constructs exhibited limited solubility. Lowering the cultivation temperature from 37°C to 18°C did not improve solubility. Inducing expression with IPTG at the early log phase significantly increased soluble CTA1-(S21P2)4-DD but not CTA1-(S14P5)4-DD. Adding non-denaturing detergents (Nonidet P40, Triton X100, or Tween 20) to the extraction buffer significantly enhanced solubility. Despite this, purification experiments yielded low amounts, only 1–2 mg/L of culture, due to substantial losses during the purification stages. These findings highlight the challenges and potential strategies for optimizing soluble expression of CTA1-DD fusion proteins for intranasal vaccines.

## Introduction

The respiratory tract is constantly exposed to the external environment and serves as a primary route for airborne pathogens to enter the body, including those causing devastating diseases such as COVID-19, SARS, MERS, influenza, and tuberculosis. It would be advantageous if these pathogens to be eliminated by the immune system in the nasal cavity before they enter

**Funding:** This research was supported by funds from National Research and Innovation Agency of Indonesia and Educational Fund Management Institution (LPDP), Ministry of Finance. RIIM 2. No 82/II.7/2022 The funders had no role in study design, data collection and analysis, decision to publish, or preparation of the manuscript." This statement has been included in the revised manuscript.

**Competing interests:** The authors have declared that no competing interests exist.

the lungs or the rest of the body. Eliciting mucosal immune responses in the nasal cavity through intranasal vaccination is a promising approach, as the nasal cavity is equipped with an advanced local lymphoid system, specifically the nasopharynx-associated lymphoid tissue (NALT) [1]. Intranasal vaccines offer several advantages over traditional injectable vaccines, including the ability to induce systemic and mucosal immunity, which is particularly effective for pathogens entering through mucosal surfaces. Additionally, intranasal administration is needle-free, enhancing patient compliance and reducing the risk of needle-associated infections and injuries [2].

Significant progress has been made in developing intranasal vaccines, particularly for influenza. Several intranasal vaccines, such as FluMist or Fluenz (AstraZeneca) and Nasovac (Serum Institute of India), have been approved for human use. These intranasal vaccines have demonstrated superiority over systemic vaccines, eliciting robust immune responses, including mucosal-neutralizing antibodies and systemic protection against homologous and heterologous viruses, without additional adjuvants [2].

Likely inspired by the success of the intranasal influenza vaccine, there is significant interest in developing intranasal vaccines for COVID-19. Various types of intranasal vaccines are being developed, including recombinant Sendai virus expressing the receptor-binding domain of SARS-CoV-2 [3], VLP-based vaccines paired with adjuvants [4], and DelNS1-nCoV-RBD live attenuated influenza vaccine [5]. These vaccines have shown efficacy in preclinical and clinical studies, with some at the stage of clinical trials demonstrating safety, immunogenicity, and potential for broader protection against emerging variants like Omicron. Intranasal vaccination presents a promising approach to combat COVID-19.

Most of the aforementioned intranasal vaccines are attenuated or adenovirus-vectored vaccines. Although these vaccines induce strong immunity, they require cold chains and often have unacceptable side effects [6]. Subunit vaccines, especially those based on epitopes, are expected to be safer. However, subunit vaccines without potent adjuvants appear ineffective in eliciting protective mucosal immune responses. Inadequate stimulation by a protein antigen may even lead to immune tolerance, which further complicates the issue. Cholera toxin is one of the most potent adjuvants for mucosal vaccination, but its high toxicity prevents its use as an adjuvant. Agren and colleagues have pioneered the elimination of the toxic nature of the protein by fusing the catalytic component of the toxin (CTA1) with a two-domain D of staphylococcal protein A (DD). The latter component enables the fusion protein to bind to B cells of all isotypes and then transform into plasma cells, producing immunoglobulins. The resulting fusion protein, CTA1-DD, retains the full adjuvanticity of cholera toxin but is entirely non-toxic [7–9]. The CTA1-DD demonstrates superior promotion of long-term immune responses compared to aluminium salts (Alum) and Ribi adjuvants [10]. An intranasal CTA1-DD-adjuvanted H3N2 split influenza vaccine elicited high titers of specific IgA in bronchoalveolar and vaginal lavages, as well as IgM and IgG in the sera of experimental animals [11]. The CTA1-DD is not only potent in inducing an IgA response but also effective in preventing the development of immune tolerance [12].

In addition to mixing antigens with CTA1-DD, protein antigens or peptides may be fused to CTA1-DD into a single fusion protein to ensure consistent adjuvanticity towards the protein antigen. An intranasal vaccine made by fusing the ectodomain matrix-2 protein (M2e) of an influenza virus between CTA1 and DD was reported to be highly effective in mice, promoting high specific serum IgG and mucosal IgA and providing strong protection against a potentially lethal challenge infection with influenza virus [13, 14]. An intranasal vaccine for human respiratory syncytial virus (hRSV) made by fusing the prefusion F protein (RBF) of the virus to the C-terminal end of CTA1-DD was reported to be effective based on vaccination and challenge trials in mice. The CTA1-DD-RBF vaccine stimulated the production of hRSV F-specific

neutralizing antibodies (IgG1, IgG2a, sIgA) and T cell immunity in mice, effectively protecting the vaccinated animals from the hRSV challenge [15].

In this study, we aim to construct fusion proteins consisting of tandem repeat epitopes S14P5 or S21P2 with CTA1 and DD, resulting in CTA1-(S14P5)4-DD and CTA1-(S21P2) 4-DD fusion proteins, which are prokaryotically expressible, functional, and soluble. Epitopes S14P5 and S21P4 are linear epitopes of the spike protein of SARS-CoV-2 that have been shown to induce neutralizing antibodies [16]. Previously, S14P5 and S21P4 have been formulated in tandem repeats and, when injected into rabbits, induced robust antibody responses that recognized the SARS-CoV-2 virus [17]. For these reasons, CTA1-(S14P5)4-DD and CTA1-(S21P2)4-DD are expected to elicit strong neutralizing antibody responses and, therefore, are suitable candidates for intranasal vaccines against COVID-19. The inclusion of linear epitopes in the vaccine design may contribute to enhanced stability, as they are less reliant on specific tertiary structures for their antigenicity compared to conformational epitopes. However, the overall thermostability of the vaccine will depend on the properties of the entire fusion protein, including its folding and interactions between domains. In this study, we seek to identify potential challenges in protein expression and purification processes that may hinder the development of these fusion proteins as vaccine candidates. By elucidating the underlying mechanisms contributing to limited solubility and low yield, our study aims to provide valuable insights for optimizing the production of CTA1-(S14P5)4-DD and CTA1-(S21P2) 4-DD, thereby enhancing their potential as viable components in future vaccine formulations. The approach used in this study should also be readily applicable to other infectious diseases.

## Materials and methods

### Construction and *in silico* analysis of CTA1-(S14P5)4-DD and CTA1-(S21P2)4-DD

The CTA1-(S14P5)4-DD and CTA1-(S21P2)4-DD constructs were generated by fusing the catalytic subunit of cholera toxin (CTA1) at the N-terminal end of the tandem repeat SARS-CoV-2 epitopes S14P5 or S21P2 and the two-domain D of the staphylococcal protein A at the C-terminal end (Fig 1A, S1 Data). The sequence for CTA1 was obtained from a previous study [18] with a modification where residue phenylalanine at residue 132 was replaced with serine to increase protein solubility [19]. The amino acid sequence of domain D staphylococcal protein A was from an earlier publication [20], and the tandem repeats S14P5 or S21P2 were from previous studies [16, 17]. A peptide linker, GGGS, was placed between CTA1 and the peptide, between the peptide and DD, and between the peptide repeats.

The characteristics of CTA1-(S14P5)4-DD and CTA1-(S21P2)4-DD were predicted based on their respective amino acid sequences. The general attributes of the recombinant proteins were assessed using the ExPASy application (https://web.expasy.org/protparam/). Solubility predictions were conducted employing DeepSoluE (http://39.100.246.211:10505/DeepSoluE/), SoluProt (https://loschmidt.chemi.muni.cz/soluprot/), and NetSolP0.1 (https://services.health tech.dtu.dk/services/NetSolP-1.0/), while propensity to aggregate was evaluated using Aggrescan (http://bioinf.uab.es/aggrescan/). Tertiary structure prediction of CTA1-(S14P5)4-DD and CTA1-(S21P2)4-DD was performed utilizing the open-source software Alphafold2 and Chimera X [21, 22].

### Expression of CTA1-(S14P5)4-DD and CTA1-(S21P2)4-DD

Synthetic genes for CTA1-(S14P5)4-DD and CTA1-(S21P2)4-DD were generated by reverse translation of the amino acid sequences and codon optimization for prokaryotic expression.

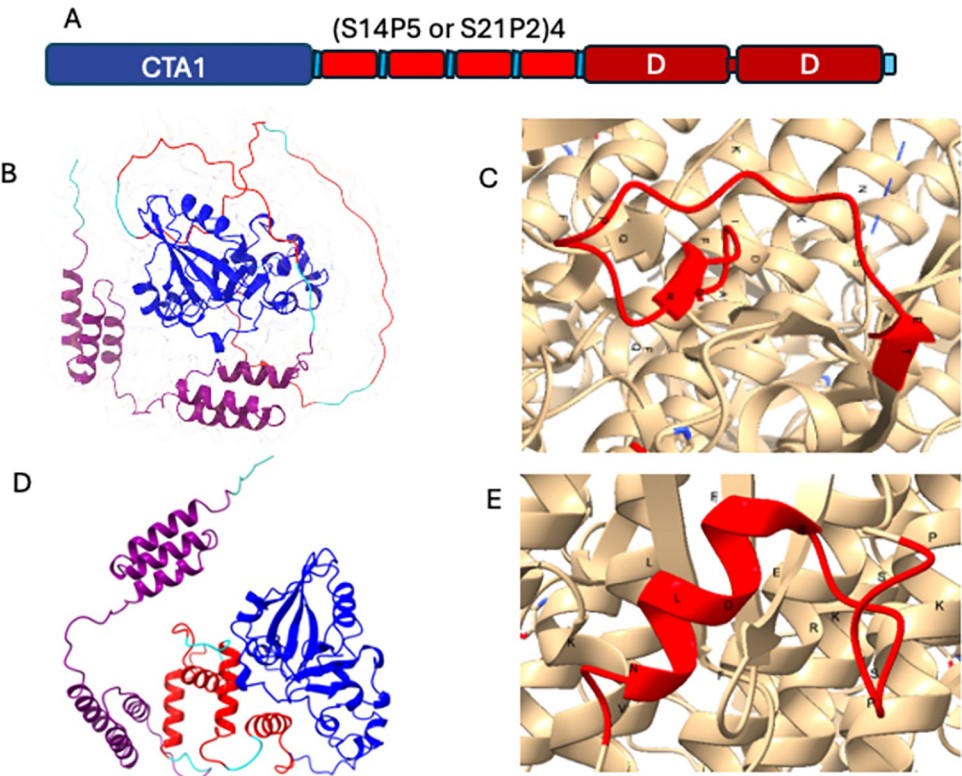

**Fig 1.** (A) Schematic representation of the fusion proteins CTA1-(S14P5)4-DD and CTA1-(S21P2)4-DD, illustrating the arrangement of the CTA1, tandem repeat epitopes, and DD domains. (B) Predicted tertiary structure of CTA1-(S14P5)4-DD by AlphaFold2, visualized using ChimeraX. CTA1 is colored in blue, (S14P5)4 in red, and DD in violet. (C) Cryo-EM structure of the SARS-CoV-2 spike protein (PDB # 6VXX), highlighting the secondary structure of the linear epitope S14P5 in red. (D) Predicted tertiary structure of CTA1-(S21P2)4-DD by AlphaFold2, visualized using ChimeraX. CTA1 is colored in blue, (S21P2)4 in red, and DD in violet. (E) Cryo-EM structure of the SARS-CoV-2 spike protein (PDB # 6VXX), highlighting the secondary structure of the linear epitope S21P2 in red.

The synthetic genes were inserted into the pET30a(+) (Novagen) expression plasmids (Genscript Inc. USA), which added a His-tag to the C-terminal end of the recombinant proteins. The plasmids were transformed into a competent BL21(DE3) strain of *E. coli* (Thermo Fisher Scientific) using a protocol described in a previous study [17]. Incubation temperatures of 37˚C and 18˚C were tested for 2 hours and 6 hours to obtain the highest amount of recombinant protein in soluble form. Induction was carried out at $OD_{600}$ values of 0.1 and 0.4. The culture medium used was Luria-Bertani (LB) broth with 30 μg/mL kanamycin, and the concentration of IPTG for induction was 0.3 m*M* in all experiments. Different solutions were tested to extract proteins from bacterial cells. Various solutions, including native buffer (0.5 *M* NaCl, 0.1 *M* sodium phosphate. pH 8), Nonidet P40, Triton X-100, and Tween-20, each at concentrations of 0.01%, 0.05%, and 0.1% in the native buffer, were tested for protein extraction. Chemicals were sourced from Sigma Aldrich.

## Analysis of total and soluble fusion proteins

To analyze total protein, 0.5 mL of bacterial culture was centrifuged at 10,000 x g for 5 minutes to pellet the cells. The resulting pellet was solubilized in 50 μL of SDS-PAGE sample buffer, and 10 μL of this suspension was loaded into a well of a 15-well polyacrylamide gel of the Mini-PROTEAN II electrophoresis apparatus (Bio-Rad). The percentage of polyacrylamide was 10% for the separating gel and 4% for the stacking gel.

To analyze soluble protein, 1 mL of bacterial culture was centrifuged at 10,000 x g for 5 minutes to pellet the cells. The pellet was resuspended in 0.5 mL of PBS containing 10 m*M* EDTA and sonicated for three intervals of 30 seconds each. Following sonication, the sample was centrifuged again at 10,000 x g for 5 minutes, and the supernatant was carefully collected. The proteins in the supernatant were precipitated with trichloroacetic acid according to the standard protocol [23]. The precipitated proteins were then solubilized in 50 μL of SDS-PAGE sample buffer, and 10 μL of this solution was loaded into an SDS-PAGE gel as previously described.

For experiments involving larger culture volumes, cell pellets from 150 mL cultures were resuspended in three separate 8 mL portions of native buffer, with and without the addition of 0.1% Nonidet P-40. The cell suspension was sonicated for five intervals of 30 seconds each, followed by centrifugation at 6,000 x g for 15 minutes, with the supernatant carefully collected. For small-scale purification, we used Mini-Bio Spin™ chromatography columns (Bio-Rad) with 100 μL Ni-NTA agarose (Qiagen GmbH) and a 0.5 mL sample volume. For larger-scale purification, we employed Econo-Pac® chromatography columns (Bio-Rad) with 1 mL Ni-NTA agarose and a 24 mL sample volume. Purification was conducted according to the vendor's protocol with some modifications. The binding and washing buffer consisted of 0.5 *M* NaCl, 0.1 *M* sodium phosphate, and 30 m*M* imidazole at pH 8 (native buffer). The elution buffer contained 0.5 *M* imidazole in the native buffer. In some experiments, detergents such as Nonidet P40, Triton X-100, or Tween-20 were added to the buffers at concentrations of 0.01%, 0.05%, and 0.1%. Dialysis was performed using 10 kDa-MWCO dialysis tubing (Sigma Aldrich) against PBS, and concentration was performed using 10 kDa-MWCO centrifugal filter units (Ultracel®-10 K, Amicon).

For the immunoblot assay, proteins from the polyacrylamide gel were transferred onto a nitrocellulose membrane, and CTA1-(S14P5)4-DD and CTA1-(S21P2)4-DD were identified respectively with rabbit-anti-(S14P5)4 and rabbit-anti-(S21P2)4 antibodies produced in our previous studies [17]. Protein concentrations were assessed by densitometry of protein bands on SDS-PAGE gel or immunoblot, using ImageJ open-source software (NIH, USA). The images of the SDS-PAGE and immunoblots were imported into ImageJ and converted to 32-bit format. Background subtraction was performed to enhance accuracy. A rectangle was drawn around the protein bands, and a plot profile of each band was generated. A line was then drawn to connect the base of each protein peak, and the area under the curves was measured. Special precautions were taken to ensure the accurate delineation of the peak bases, thus providing actual peak areas. These peak areas correspond to the grayscale intensity of the bands, which directly correlates with the amount of protein present. The absolute concentrations of proteins were determined using a standard curve constructed with Bovine Serum Albumin (BSA) (Sigma Aldrich) at concentrations of 0, 0.31, 0.62, 1.25, 2.5, and 5.0 μg/mL, which were loaded on the same gel.

## Statistical analysis

Descriptive statistics were used to summarize the data. The Wilcoxon rank-sum test was employed to compare protein concentrations between different experimental conditions. This non-parametric test is suitable for small sample sizes or data that are not normally distributed. A *p*-value less than 0.05 was considered statistically significant. All statistical analyses were done using Jamovi, an open-source statistical software (https://www.jamovi.org).

## Result

### *In silico* analysis of CTA1-(S14P5)4-DD and CTA1-(S21P2)4

The general characteristics of CTA1-(S14P5)4-DD and CTA1-(S21P2)4-DD, including the number of amino acids, molecular weight, aliphatic index, grand average of hydropathicity

**Table 1. General or predicted characteristics of CTA1-(S14P5)4-DD, CTA1-(S21P2)4-DD, and Bovine Serum Albumin (BSA).**

| Characteristics | CTA1-(S14P5)4-DD | CTA1-(S21P2)4-DD | BSA |
|---|---|---|---|
| No. amino acids[1] | 419 | 423 | 590 |
| Molecular weight[1] | 46.5 kDa | 46.9 kDa | 67.3 kDa |
| Isoelectric point (pI) [1] | 5.97 | 6.35 | 5.76 |
| Aliphatic index[1] | 55.73 | 60.69 | 75.73 |
| Grand average of hydropathicity (GRAVY)[1] | -0.910 | -0.866 | -0.482 |
| Instability index[1] | 53.6 | 59.3 | 40.91 |
| Solubility prediction (DeepSoluE[2]) | 0.9513 | 0.9647 | 0.8553 |
| Number of Hot Spots (nHS) (Aggrescan[3]) | 6 | 10 | 17 |
| Normalized nHS for 100 residues (NnHS) (Aggrescan[3]) | 1.432 | 2.364 | 2.881 |
| Solubility prediction of protein over-expressed in *E. coli* | | | |
| SoluPro[4] | 0.858 | 0.887 | not relevant |
| NetSolP[5] | 0.5975 | 0.5946 | not relevant |

Note

[1] https://web.expasy.org/protparam/

[2] http://39.100.246.211:10505/DeepSoluE/

[3] http://bioinf.uab.es/aggrescan/

[4] https://loschmidt.chemi.muni.cz/soluprot/

[5] https://services.healthtech.dtu.dk/services/NetSolP-1.0/

(GRAVY), and instability index, are presented in Table 1. To facilitate a better understanding of the proteins, we included the well-known protein bovine serum albumin (BSA) for comparison, whose amino acid sequence was obtained from GenBank (accession number AAA51411.1). Both CTA1-(S14P5)4-DD and CTA1-(S21P2)4-DD have molecular weights of approximately 47 kDa, are acidic (pI<7), and more hydrophilic than BSA, as indicated by their lower GRAVY scores. According to the instability index, both proteins were less stable than BSA. The solubility of CTA1-(S14P5)4-DD and CTA1-(S21P4)4-DD were predicted to be higher than the highly soluble BSA. This higher solubility was in agreement with the higher hydrophilicity. Moreover, the CTA1-(S14P5)4-DD and CTA1-(S21P2)4-DD were predicted to have a lower propensity for aggregation than BSA, as indicated by the higher number of hot spot area (nHS) and normalized nHS for 100 residues (NnHS).

Upon overexpression in *E. coli*, CTA1-(S14P5)4-DD and CTA1-(S21P2)4-DD were predicted to exhibit high solubility using online computational tools. The SoluPro application generated high prediction values, precisely 0.858 for CTA1-(S14P5)4-DD and 0.887 for CTA1-(S21P2)4-DD. In contrast, the NetSolP application yielded considerably lower predictions of only 0.6 for both proteins.

The schematic and tertiary structures of CTA1-(S14P5)4-DD and (S21P2)4 are shown in Fig 1. The schematic representation provides an overview of these constructs' domain organization and sequences (Fig 1A). The CTA1 domain consists of two groups of three antiparallel beta sheets, several short alpha helices, and connecting loops. The DD component comprised a pair of three long, parallel alpha helices connected by a long coil. The (S14P5)4 appeared as a long coil, whereas the (S21P2)4 appeared as four alpha helices with coils at both ends (Fig 1B and 1D). The structures of S14P5 and S21P2 in the predicted structure of CTA1-(S14P5)4-DD and (S21P2)4 were similar to those in the cryo-electron microscopy structure of SARS-CoV2 spike protein previously submitted to the Protein Data Bank ((PDB # 6VXX) (Fig 1C and 1E).

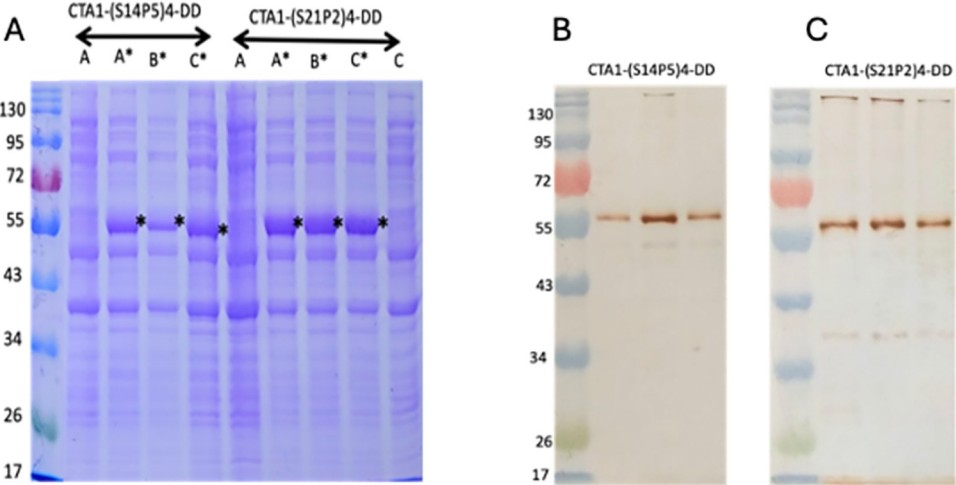

**Fig 2. Total proteins of *E. coli* transformed with expression plasmid carrying either *CTA1-(S14P5)4-DD* or *CTA1-(S21P2)4-DD* genes.** A. Three colonies grown in kanamycin-supplemented LB broth were harvested before (A, B, and C) and after induction with IPTG (A', B' and C'). B and C: The *CTA1-(S14P5)4-DD-* and *CTA1-(S21P2)4-DD-* transformed *E. coli* were probed with relevant specific antibodies.

## Expression of CTA1-(S14P5)4-DD and CTA1-(S14P5)4-DD

The *E. coli* cells transformed with *the CTA1-(S14P5)4-DD* or *CTA1-(S21P2)4-DD* constructs exhibited robust protein expression upon induction with IPTG (Fig 2A). No protein expression was observed before IPTG induction, indicating correct regulation of genetic constructs and a responsive expression system. Substantial expression levels were evidenced by the intense protein bands observed in the gels. Densitometry analysis determined the expression levels of CTA1-(S14P5)4-DD and CTA1-(S21P2)4-DD to be 30.5% and 45.8% of total cell protein, respectively. Immunoblot analysis using monospecific anti-(S14P5)4 and anti-(S21P2)4 antibodies confirmed the identity of the expressed proteins (Fig 2B and 2C), validating their expected characteristics. The molecular weights of CTA1-(S14P5)4-DD and CTA1-(S21P2) 4-DD were determined to be 61.2 kDa and 61.4 kDa, respectively, significantly higher than the calculated molecular weights based on their amino acid compositions, which were 46.5 kDa and 46.9 kDa, respectively (Table 1).

## Effect of growth stage ($OD_{600}$) at induction, temperature and duration of incubation

Contrary to the *in silico* prediction, both CTA1-(S14P5)4-DD and CTA1-(S21P2)4-DD were poorly soluble when overexpressed in *E. coli*. The soluble fractions constituted only a negligible portion of the total proteins, and the exact quantification was often unreliable due to smearing observed in many protein bands (S1 Fig, S1 Table). Lowering the culture temperatures from 37˚C to 18˚C did not increase protein solubility ($p > 0.05$). Quantitative data showed that lowering the culture temperature from 37˚C to 18˚C decreased the soluble protein yield for CTA1-(S14P5)4-DD from 10% to 5% and for CTA1-(S21P2)4-DD from 12% to 6%. Prolonging incubation from 3 to 6 hours increased the amount of soluble proteins ($p < 0.05$), with CTA1-(S14P5)4-DD rising from 8% to 15% and CTA1-(S21P2)4-DD from 9% to 18%. Additionally, induction at an early log phase ($OD_{600}$ of 0.1) resulted in higher production of soluble CTA1-(S21P2)4-DD (17%) compared to an $OD_{600}$ of 0.4 (12%). However, a significant increase by induction at the early log phase was not observed in the production of soluble CTA1-(S14P5)4-DD ($p > 0.05$) (S2 and S3 Tables).

### Effect of detergent in the extraction and purification buffer

Protein purification using the native buffer (0.5 $M$ NaCl, 0.1 $M$ sodium phosphate, pH 8) with the Ni-NTA purification system yielded low amounts of CTA1-(S14P5)4-DD and even lower amounts of CTA1-(S21P2)4-DD (Fig 3). Adding detergents to the native buffer significantly increased the yields. For CTA1-(S14P5)4-DD, Nonidet P40 at 0.01% increased the yields by 5 times, at 0.05% and 0.1% by 7 times over the native buffer. Triton X-100 at 0.01%, 0.05%, and 0.1% increased the yields by 5, 6, and 7 times, respectively. Tween 20 was less effective, with 0.01% yielding a 2 times increase, and either 0.05% or 0.1% yielding a 5 times increase over the native buffer.

The increase in soluble yields was more profound with CTA1-(S21P2)4-DD. Nonidet P40 at 0.01% increased yields by 7 times, at 0.05% by 13 times, and at 0.1% by 16 times over the native buffer. Triton X-100 at 0.01%, 0.05%, and 0.1% increased yields by 8, 11, and 8 times, respectively. Tween 20 at 0.01% yielded a 2-fold increase, and at either 0.05% or 0.1% yielded an 11-fold increase over the native buffer (Fig 3). Statistical analysis was not carried out because there was no replication of the samples; nevertheless, the profound increase in the yields of soluble proteins by adding the detergents should indicate their significance.

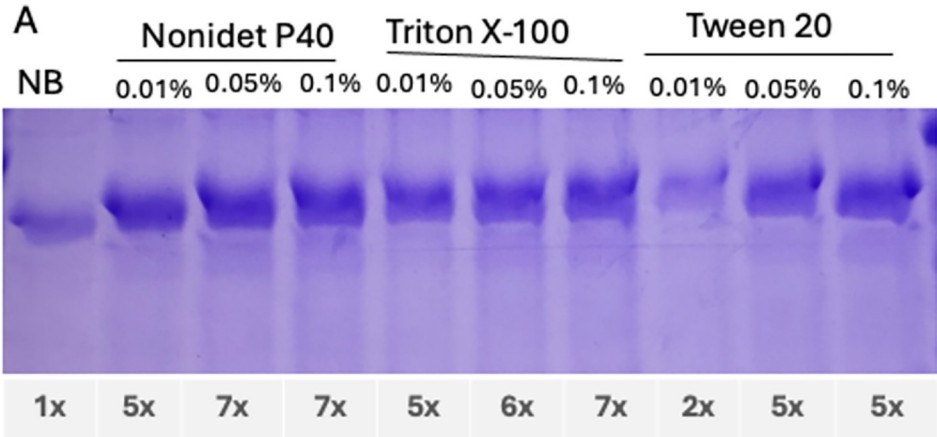

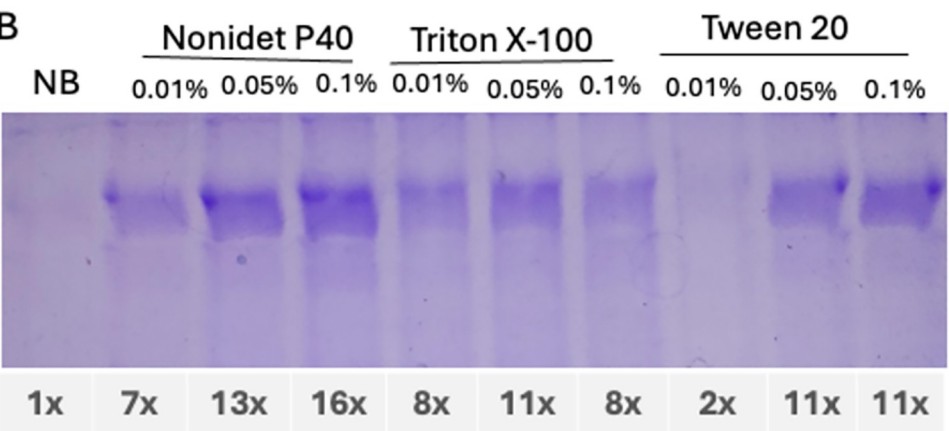

**Fig 3. Effect of detergents on the yield of proteins purified using NiNTA column.**

Further yield improvements were observed with higher concentrations of detergents, including Tween 20 at 0.05% and 0.1%. However, increasing the concentration from 0.05% to 0.1% did not significantly increase in soluble protein amounts. Therefore, either 0.05% or 0.1% concentrations was considered adequate, with no marked difference in the yield of soluble proteins between these concentrations.

## Effectivity of purification of CTA1-(S14P5)4-DD and CTA1-(S14P5)4-DD

The efficacy of purifying CTA1-(S14P5)4-DD and CTA1-(S21P2)4-DD was tested under the optimized conditions determined in this study: 37°C culture temperature and 0.1% Nonidet-P40 extraction buffer, using a larger (150 mL) culture volume. The results of these experiments are detailed in Tables 2 and 3 and S2 Fig. After a 2-hour induction period, the bacterial cells contained 24–42 mg/L of CTA1-(S14P5)4-DD and 32–40 mg/L of CTA1-(S21P2)4-DD, based on densitometry using BSA as the standard. Despite the substantial initial protein amounts, the yields were low at the end of the purification process. Following elution, dialysis, and concentration in buffers without detergent, the yields were less than 5% of the total protein in the bacterial cells for both proteins. Adding Nonidet P-40 to the elution and dialysis buffers only marginally increased the yields to 2.4 mg/L of culture for CTA1-(S14P5)4-DD and 1.15 mg/L for CTA1-(S21P2)4-DD. These yields contrasted with previous small-scale (1.5 mL culture) experiments, where the addition of Nonidet P-40 to the extraction buffer led to a sevenfold increase in the yield of CTA1-(S14P5)4-DD and a sixteenfold increase in the yield of CTA1-(S21P2)4-DD (Fig 3). The discrepancy in the yields between the small and larger volumes can be attributed to several factors. At small scales, higher mixing efficiency during bacterial cultivation, more efficient cell sonication, and much shorter purification times were achieved. In contrast, larger-scale processes face challenges in maintaining consistent mixing, efficient sonication, and timely purification, leading to variations in protein yield and purity. Further optimization of these parameters is required to improve the efficiency and scalability of large-scale purifications.

**Table 2. Purification of CTA1-(S14P5)4-DD at different stages of the purification process, comparing elution without Nonidet-P40 and with Nonidet-P40.**

| Purification stage | Eluant without Nonidet-P40 | | | Eluant containing Nonidet P-40 | | |
|---|---|---|---|---|---|---|
| | Protein (µg) in the band | Protein µg/mL culture | Recovery | Protein (µg) in the band | Protein µg/mL culture | Recovery |
| Original cells (~0.05 mL culture) | 1.2 | 24 | 100% | 2.1 | 42 | 100% |
| Cells lysate (~0.125 mL culture) | 1.9 | 15.2 | 63.3% | 5.5 | 44 | 105% |
| Eluate (~0.5 mL culture) | 0.6 | 1.7 | 7.1% | 1.0 | 2 | 4.8% |
| Dialysed and concentrated eluate (2 mL) | 1.1 | 0.55 | 4.2% | 4.8 | 2.4 | 5.7% |

**Table 3. Purification of CTA1-(S21P2)4-DD at different stages of the purification process, comparing elution without Nonidet-P40 and with Nonidet-P40.**

| Purification stage | Eluant without Nonidet-P40 | | | Eluant containing Nonidet P-40 | | |
|---|---|---|---|---|---|---|
| | Protein (µg) in the band | Protein µg/mL culture | Recovery | Protein (µg) in the band | Protein µg/mL culture | Recovery |
| Original cells (~0.05 mL culture) | 2 | 40 | 100% | 1.6 | 32 | 100% |
| Cells lysate (~0.125 mL culture) | 2.5 | 20 | 50% | 3.5 | 28 | 88.5% |
| Eluate (0.5 mL culture) | 1.3 | 2.6 | 6.5% | 1.2 | 2.4 | 7.5% |
| Dialysed and concentrated eluate (~2 mL culture) | 1.9 | 0.95 | 2.4% | 2.3 | 1.15 | 3.6% |

The most significant protein losses occurred during the NiNTA column separation stage, with over 90% of the target protein being lost during this process. During the lysate preparation stage, there was only a slight reduction, likely due to overestimated protein quantification caused by target band smearing, possibly indicating contamination with bacterial nucleic acids resulting from sonication (S2 Fig, lane 2). Further substantial losses occurred during concentration and desalting in PBS, with more than 50% of the target proteins present in the eluate being lost. Adding Nonidet-P40 to the elution increased recovery, although significant losses could not be prevented.

## Discussion

In the present study, we constructed two fusion proteins, CTA1-(S14P5)4-DD and CTA1-(S21P2)4-DD, as candidates for intranasal vaccines against SARS CoV-2. Each protein contains a linear epitope S14P5 or S21P2 of SARS CoV-2 spike protein, which is efficacious in inducing neutralizing antibodies [16]. The linear epitopes were made in the form of four tandem repeats. Our previous study confirmed that the S14P5 and S21P2 in this form enhanced the immunogenicity of the epitopes, and the induced antibodies effectively recognized the SARS-CoV-2 virus [17]. To further enhance the immune responses and to optimize them as intranasal vaccines, each of the tandem repeat epitopes has been fused to the catalytic subunit of cholera toxin (CTA1), and the domain D of staphylococcal protein A in dimer form (DD). The choice of domain D, instead of other domains, was beneficial due to its ability to bind not only IgG but also IgA and IgM [20].

*In silico* analysis of protein structure and function has become indispensable in modern biomedical research, particularly with the rapid advancement of bioinformatics tools. These computational methods can predict protein behavior, provide insights into protein interactions, stability, and potential binding sites, and guide the design of experiments, thus saving time and resources. To assess the functionality of the components within the fusion proteins, CTA1-(S14P5)4-DD and CTA1-(S21P2)4-DD, we performed *in silico* analyses to predict their compatibility with their native structures. The three-dimensional structures of the fusion proteins were predicted using AlphaFold-2, a computational method that predicts 3D protein structures from their respective amino acid sequences with near-experimental accuracy [22].

The structure of the CTA1-(S14P5)4-DD and CTA1-(S14P5)4-DD components, as predicted by AlphaFold-2, were compatible with their original structure, suggesting the proper function of the proteins. The CTA1, which is the catalytic, ADP-ribosyltransferase and a NAD-glycohydrolase domain of cholera toxin, can be delineated into three discrete regions: $CTA1_1$ (residues 1–132), $CTA1_2$ (residues 133 to 161), and $CTA1_3$ (residues 162–193). The $CTA1_1$ forms a compact globular unit characterized by a combination of alpha-helices and beta-strands, harboring a catalytic cleft presumed to be the site of NAD and substrate binding. The $CTA1_2$ acts as a flexible linker bridging $CTA1_1$ and $CTA1_3$. The $CTA1_3$ is marked by a dense arrangement of hydrophobic residues [18]. Our predicted molecular model of CTA1-(S14P5)4-DD and CTA1-(S21P2)4-DD aligns well with the established crystallographic structure. Specifically, residues 1–132 exhibit a similar secondary structure composition of alpha-helices and beta-strands, suggesting the presence of the $CTA1_1$ subunit. Residues 133–161 manifest as an elongated coil, consistent with the characteristics of $CTA1_2$, while residues 162–193 adopt a conformation resembling a tangled loop interspersed with short alpha helices, indicative of $CTA1_3$.

The DD component of the proteins appeared as a pair of three long, parallel alpha helices. This structure is compatible with the crystal structure domain D of staphylococcal protein A previously reported [20]. The secondary structure of epitopes S14P5, which appeared as a long

loop, and S21P2, which appeared as an alpha helix and loops, were compatible with the relevant segment in the crystal structure of spike protein SARS CoV2 deposited in the Protein Data Bank [24].

Proper design constitutes the initial step in obtaining CTA1-(S14P5)4-DD and CTA1-(S21P2)4-DD that function as intended. The subsequent crucial step involves the expression and purification of the constructed fusion proteins, ensuring the preservation of their structure and function. Prokaryotic expression remains the preferred choice for protein expression due to its simplicity and cost-effectiveness. However, a significant challenge in prokaryotic expression is the formation of proteins in insoluble aggregates [25, 26]. Expressing proteins in soluble forms within *E. coli* offers notable advantages. Soluble expression guarantees the maintenance of proteins' native conformation, thereby preserving their functionality. In contrast, expression in inclusion bodies often requires denaturation and subsequent renaturation steps, posing risks of protein function loss. Moreover, isolating proteins from insoluble inclusion bodies entails higher costs and longer processing times [27].

Predicting protein solubility before initiating expression confers significant advantages, particularly for engineered constructs like CTA1-(S14P5)4-DD and CTA1-(S21P2)4-DD. Therefore, it was imperative to evaluate their solubility, primarily upon overexpression in *E. coli*. However, current bioinformatic tools still exhibit suboptimal reliability in predicting the solubility of overexpressed proteins, as evident from this study. Based on their amino acid sequences, both CTA1-(S14P5)4-DD and CTA1-(S21P2)4-DD demonstrated high solubility. These proteins displayed a notably higher probability of solubility than bovine serum albumin, a protein renowned for its solubility. Although highly soluble proteins are not necessarily soluble when over-expressed in *E. coli*, this is true for both CTA1-(S14P5)4-DD and CTA1-(S21P2)4-DD, which were predicted to be soluble [28].

The CTA1-(S14P5)4-DD and CTA1-(S21P2)4-DD were predicted to be highly soluble on overexpression in *E. coli* by two computational tools: (1) SoluPro, which predicts solubility based on amino acid composition, dipeptide composition, and physicochemical properties, employing a machine learning approach, and (2) NetSolP, which predicts solubility based on amino acid composition, secondary structure, and solvent accessibility employing a neural network-based approach. These tools have undergone significant improvements, achieving accuracies of 74% for SOLUPro and 70% for NetSolP, surpassing many previous methods [29, 30]. However, despite these advancements, there remains a risk of misprediction, particularly for engineered proteins like CTA1-(S14P5)4-DD and CTA1-(S21P2)4-DD, as these computational tools may not have been trained on similar proteins. Furthermore, *in silico* tools often do not account for complex cellular environments, such as expression levels, folding mechanisms, chaperone interactions, and post-translational modifications, which can significantly influence solubility [31, 32]. The aggregation propensity, a critical factor in protein solubility, is also challenging to predict accurately [33, 34]. Future computational tools should incorporate more diverse training datasets and consider the cellular context and experimental conditions to improve solubility predictions.

In experiments aimed at optimizing the solubility of expressed protein in *E. coli*, quantifying expressed protein is challenging. In the present study, we used densitometry techniques to quantify the proteins in SDS PAGE gels or immunoblots with specific antibodies. This technique can measure both soluble and insoluble proteins. Additionally, it can be used to measure proteins in mixtures with other unrelated proteins present in the samples. Despite these advantages, the technique requires high-resolution SDS PAGE gels or immunoblots that allow for the complete separation of proteins, which is often unattainable.

The formation of insoluble expressed proteins as inclusion bodies emerged as one of the most significant obstacles in prokaryotic expression systems. Inclusion bodies are formed due

to a rapid rate of protein expression, a key objective in recombinant protein production. When protein expression exceeds the host cells' capacity for post-translational modifications and folding, misfolding occurs, leading to aggregation into inclusion bodies as hydrophobic residues become exposed. Slowing down the expression rate by modifying culture conditions, such as lowering culture temperature and the concentration of induction agents, comprise the most practical approach to increasing the solubility of expressed proteins. Lowering temperature also diminishes the hydrophobic interaction between the expressed proteins, thus reducing aggregation [26].

Attempts to convert expressed proteins in *E. coli* from insoluble inclusion bodies to soluble forms by lowering the cultivation temperature have proven successful in many proteins. For instance, a three-fold increase in the soluble fraction of green fluorescent protein (GFP) was achieved by lowering the temperature from 37˚C to 16˚C [35]. Similarly, human interferon-α2 and γ, which formed insoluble aggregates when expressed in *E. coli* at 37˚C, demonstrated increased solubility by 30–90% when cells were cultivated at 23˚C [36]. The increased solubility of heterologous proteins expressed in *E. coli* at lower temperatures has also been observed in various other proteins, including β-lactamase, human epidermal growth factor, human hemoglobin, and β-galactosidase [37–39]. However, the effectiveness of lowering the culture temperature in increasing protein solubility varies among different proteins [26]. As demonstrated in this study, lowering the cultivation temperature to 18˚C instead of the typical 37˚C did not yield the expected increase in solubility.

Moreover, in a previous study, induction of *P5βR*-transformed *E. coli* with IPTG during the early-log phase substantially increased the solubility of the expressed P5βR (progesterone 5β-reductase [40]. The increase in the solubility of expressed proteins by IPTG induction at the early log phase is likely dependent on the protein. In the present study, soluble CTA1-(S21P2)4-DD expression significantly increased, but not CTA1-(S14P5)4-DD. It is unknown why the solubility of some expressed proteins increases and not others.

Various studies have extensively explored the production of fusion proteins, CTA1-DD or CTA1-peptide-DD, utilizing *E. coli* expression systems. Despite numerous attempts, the expression of CTA1-DD or CTA1-peptide-DD consistently yields insoluble inclusion bodies, with no reported instances of soluble expression. This phenomenon has been observed across different *E. coli* expression systems, including the pUC19 expression vector with TG-1 *E. coli* in 2YT medium [8, 19, 41–43], DH5 *E. coli* in SYPPG medium [13], as well as the use of pET28a with BL-21 *E. coli* strains [15]. Despite forming insoluble inclusion bodies, the extracted proteins have been successfully purified using guanidine HCl and chromatography techniques, followed by subsequent renaturation steps. Remarkably, the purified proteins have exhibited functionality despite these initial hurdles, as evidenced by various assays. Specifically, the ADP-ribosyltransferase activity of CTA1 has been validated through NAD: agmatine assay [7, 19, 41, 43], while the DD component has been demonstrated to bind specifically to immunoglobulin receptors on B cells [7].

There are several other alternative strategies that can be exploited to improve the solubility and yield of CTA1-(S14P5)4-DD and CTA1-(S21P2)4-DD fusion proteins. Practical approaches include incorporating chemical additives like arginine, glycerol, betaine, and trehalose in the growth medium or purification buffers, which can stabilize proteins and inhibit aggregation [44]. Additionally, co-expressing molecular chaperones such as GroEL/GroES, DnaK/DnaJ/GrpE, or trigger factors can assist in proper protein folding and prevent aggregation [45].

In previous studies, purification of the fusion protein CTA1-DD expressed in *E. coli* as inclusion bodies involved solubilization with guanidine-HCl, isolation using an IgG-Sepharose-affinity column, and renaturation, resulting in yields of 8–60 mg/L culture [19, 46].

Although it is inappropriate to compare different proteins in this aspect of purification directly, the yields of CTA1-(S14P5)-DD and CTA1-(S21P2)-DD using native techniques were significantly lower. Despite efforts to enhance solubility in the present study, purification using native techniques yielded only 1–2 mg of pure proteins/L of culture.

The low yield of native CTA1-(S14P5)-DD and CTA1-(S21P2)-DD was due to substantial losses during the purification stages. Higher yields are likely achievable by optimizing the purification process, as the expressed fusion proteins constitute 30–45% of the total bacterial protein after just 2 hours of induction. Additionally, as found in this study, the inclusion of non-denaturing detergents such as Nonidet P40, Triton X100, or Tween 20 in the extraction buffer effectively solubilizes the expressed CTA1-(S14P5)-DD and CTA1-(S21P2)-DD.

The approach used in this study for intranasal vaccine development can be easily adapted to other infectious diseases. For diseases that have existed for some time, lists of linear epitopes that evoke neutralizing antibodies are likely to have been identified. Fusion proteins consisting of these tandems repeat epitopes, and CTA1-DD can be constructed for mucosal vaccines. For new emerging infectious diseases, such as those that may arise in the future, the complete sequence of the causal agents, similar to what occurred with SARS-CoV-2, will be available immediately. Linear epitopes can be identified from protein sequences using computational tools, which are becoming increasingly accurate in their predictions. As a result, vaccines could be available immediately before the disease spreads to more expansive areas.

In summary, two fusion proteins, CTA1-(S14P5)4-DD and CTA1-(S21P2)4-DD, have been constructed as candidates for intranasal vaccines against COVID-19. The constructs were expressed abundantly in *E. coli* as insoluble inclusion bodies. The solubility of the expressed proteins did not increase by lowering the cultivation temperature. Expression of soluble CTA1-(S21P2)4-DD, but not CTA1-(S14P5)4-DD, increased when the culture was induced with IPTG at the early log-phase growth. The solubility of both CTA1-(S14P5)4-DD and CTA1-(S21P2)4-DD was significantly enhanced by adding non-denaturing detergents to the extraction buffer. Several critical steps are necessary to advance these candidates toward vaccine development. First, in vivo immunogenicity and efficacy studies need to be conducted in appropriate animal models. Second, optimizing the expression and purification protocols to increase yield and solubility will be essential. Finally, detailed safety and toxicity assessments are required to ensure the candidates are safe for human use."

## Supporting information

### S1 Data. Amino-acid sequence for CTA1-(S14P5)4-DD and CTA1-(S21P2)4-DD.
(DOCX)

**S1 Fig.** Visualization of CTA1-(S14P5)4-DD (A) and CTA1(S21P2)4-DD (B) proteins using specific antibodies in immunoblot assays. Proteins were analyzed as either soluble fractions or total proteins within bacterial cells. Cultures were grown at 37°C or 18°C, incubated for 3 or 6 hours, and induced with IPTG at early log-phase ($OD_{600}$ of 0.1) or mid-log-phase ($OD_{600}$ of 0.4). Note that the samples for the soluble fractions were derived from four times more bacterial cells than those for total protein.
(TIF)

**S2 Fig. Purification of CTA1-(S14P5)4-DD and CTA1-(S21P2)4-DD, each from 150 mL cultures.** Bacterial cell lysates were prepared in PBS containing 0.1% Nonidet P40 and subjected to NiNTA column chromatography. Proteins adsorbed by the column were eluted with either 0.5 M imidazole (A, C) or 0.5 M imidazole containing 0.1% Nonidet P40. Lane 1: Proteins from 50 μL culture after induction. Lane 2: Proteins from the supernatant of cell lysate

from 125 μL culture. Lane 3: Eluate from the NiNTA column derived from 500 μL culture. Lane 4: Dialyzed and concentrated eluate from 2 mL culture.
(TIF)

**S1 Table. Percentage of soluble protein recombinants at different incubation temperatures, durations of incubation, and growth stages of IPTG induction.**
(DOCX)

**S2 Table. Production of soluble CTA1-(S14P5)4-DD, as indicated by different cultivation temperatures, incubation times, and growth stages of induction.**
(DOCX)

**S3 Table. Production of soluble CTA1-(S21P2)4-DD, as indicated by at different cultivation temperatures, incubation times, and growth stages of induction.**
(DOCX)

## Acknowledgments

The authors would like to express their sincere gratitude to Dr. Fx. Sudirman, Director of Biotis Pharmaceuticals Indonesia, for his invaluable support in the publication of this study.

## Author Contributions

**Conceptualization:** Simson Tarigan.

**Data curation:** Simson Tarigan, Gita Sekarmila,  Apas,  Sumarningsih.

**Formal analysis:** Simson Tarigan, Ronald Tarigan.

**Funding acquisition:** Simson Tarigan, Damai Ria Setyawati.

**Investigation:** Simson Tarigan.

**Methodology:** Simson Tarigan.

**Project administration:** Simson Tarigan,  Sumarningsih, Riyandini Putri.

**Resources:** Simson Tarigan.

**Supervision:** Simson Tarigan.

**Validation:** Simson Tarigan.

**Visualization:** Simson Tarigan.

**Writing – original draft:** Simson Tarigan,  Sumarningsih, Ronald Tarigan, Riyandini Putri, Damai Ria Setyawati.

**Writing – review & editing:** Simson Tarigan.

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
