## [Decision Letter · Decision Letter 0]

1 Jul 2024

PONE-D-24-23892Optimization of Soluble Expression of CTA1-(S14P5)4-DD and CTA1-(S21P2)4-DD Fusion Proteins as Candidates for COVID-19 Intranasal VaccinesPLOS ONE

Dear Dr. Tarigan,

Thank you for submitting your manuscript to PLOS ONE. After careful consideration, we feel that it has merit but does not fully meet PLOS ONE’s publication criteria as it currently stands. Therefore, we invite you to submit a revised version of the manuscript that addresses the points raised during the review process.

We look forward to receiving your revised manuscript.

Kind regards,

Haitham Mohamed Amer, PhD

Academic Editor

PLOS ONE

Journal Requirements:

"This research was supported by funds from National Research and Innovation Agency of Indonesia and Educational Fund Management Institution (LPDP), Ministry of Finance. RIIM 2. No 82/II.7/2022"

3. Please upload a new copy of Figure 2 and Supporting Figure 1 as the detail is not clear. Please follow the link for more information: 

https://blogs.plos.org/plos/2019/06/looking-good-tips-for-creating-your-plos-figures-graphics/

https://blogs.plos.org/plos/2019/06/looking-good-tips-for-creating-your-plos-figures-graphics/

Reviewers' comments:

Reviewer's Responses to Questions

**Comments to the Author**

1. Is the manuscript technically sound, and do the data support the conclusions?

Reviewer #1: Yes

Reviewer #2: Partly

2. Has the statistical analysis been performed appropriately and rigorously? 

Reviewer #1: N/A

Reviewer #2: Yes

3. Have the authors made all data underlying the findings in their manuscript fully available?

Reviewer #1: Yes

Reviewer #2: Yes

4. Is the manuscript presented in an intelligible fashion and written in standard English?

Reviewer #1: No

Reviewer #2: Yes

5. Review Comments to the Author

Reviewer #1: The manuscript "Optimization of Soluble Expression of CTA1-(S14P5)4-DD and CTA1-(S21P2)4-DD Fusion Proteins as Candidates for COVID-19 Intranasal Vaccines" by Tarigan et al describes the design, expression, and initial purification attempts of two fusion proteins intended as candidates for COVID-19 intranasal vaccines. The authors constructed fusion proteins combining the catalytic subunit of cholera toxin (CTA1), tandem repeats of SARS-CoV-2 spike protein epitopes, and a dimer of staphylococcal protein A domain D. The work provides insights into the challenges of expressing and purifying these novel constructs.

The abstract provides a comprehensive overview of the study but could benefit from being more concise. It currently contains excessive background information that dilutes the focus on the key findings. A more structured approach highlighting the objectives, methods, primary results, and main conclusions would improve its effectiveness.

The introduction effectively establishes the rationale for developing intranasal vaccines against COVID-19. It provides a good overview of the advantages of intranasal vaccination and the potential of CTA1-DD fusion proteins as mucosal adjuvants. The authors successfully contextualize their work within the broader field of vaccine development. However, the transition to the specific aims of this study could be more clearly articulated.

Materials and Methods section is generally well-organized, detailing the in silico analyses and experimental procedures. However, some critical information is missing:

1. The exact sequences of the constructs are not provided, which is essential for reproducibility.

2. The specific E. coli strain used for expression is not mentioned.

3. The protein quantification method using densitometry needs more detailed description.

4. Statistical analysis methods are mentioned but not fully explained.

The results are presented in a logical sequence, starting with in silico predictions and moving through expression optimization attempts. The authors provide details on their experimental findings. However, the section would benefit from:

1. More quantitative data presentation, particularly in comparing soluble protein yields under different conditions.

2. Clearer explanation of the discrepancies between small-scale and large-scale purification results.

3. More consistent application of statistical analyses to support the conclusions drawn.

The figures are generally informative, but some, particularly those showing gel images, could be improved in terms of clarity and labeling.

The discussion effectively contextualizes the results within the broader literature on fusion protein expression. The authors are commendably candid about the limitations encountered in their work. However, the section could be strengthened by:

1. More critical analysis of why the in silico predictions of solubility were inaccurate.

2. Deeper exploration of alternative strategies to improve solubility and yield.

3. Clearer articulation of the next steps needed to advance these candidates toward vaccine development.

4. Discussion of the potential immunogenicity and protective efficacy of these constructs, even if not experimentally addressed in this study.

Authors cited some useful references demonstrating a good grasp of the relevant literature. However, some additional recent papers on intranasal COVID-19 vaccine development could be included to further contextualize this work.

Overall, this manuscript presents valuable initial work on novel fusion protein constructs for potential COVID-19 intranasal vaccines. The manuscript would benefit from a thorough proofreading to correct minor grammatical errors and improve clarity in some sections. With these improvements, this work could make a significant contribution to the ongoing efforts in developing effective mucosal vaccines against COVID-19.

Reviewer #2: The manuscript describes the construction, in silico prediction of 3D structures, cloning, and attempts of production and purification of two fusion proteins, CTA1-(S14P5)4-DD and CTA1-(S21P2)4-DD, that are candidates to new intranasal vaccines for SARS-CoV-2. In general, the text is well-written, and the results are presented in a clear manner, but there are some important points that the authors should consider before publishing the work, including conceptual errors and poor description of Material and Methods.

First, the title is not very precise, as in fact no optimization was really achieved.

Although the abstract and title bring the conceptually correct term “fusion protein”, on pages 3 (L. 89-99) and 4 (L. 119) the authors use conjugation or conjugating. Normally, in the vaccine field, conjugation refers to a chemical reaction to bind two molecules and is largely applied to conjugate vaccines, in which polysaccharides are chemically bound to a carrier protein. Therefore, the terms conjugation or conjugating should be replaced by fusion or fusing (also fused) to avoid misunderstandings. Also, it is misleading to call a 150 mL culture (information found only in Fig S2 caption) a scale-up experiments, since the term “scale-up” brings at least the idea of bioreactor cultivation.

The authors affirm that “the vaccine candidates are expected be thermostable because they are based on linear epitopes” (p. 4, L. 108-109). This statement related two independent facts without scientific basis, there is no reason for linear epitopes be thermostable or conformational epitopes be labile. The stability of a protein is determined by a series of physical-chemical characteristics and presence of protease cleavage sites rather than if they are composed of linear epitopes.

The authors claim that the approach of this study could be readily applied to other infectious disease (p. 4, L. 114-115), using linear epitopes that induce neutralizing antibodies (p. 15, L. 437-445). While it might be true that one can use bioinformatics to design similar fusion molecules simply changing the epitopes, as each novel protein will present distinct physical-chemical characteristics that will influence the production and purification processes, all new constructs will need a completely new process development to really generate a vaccine candidate. Even from the immunological point of view, it is hard to say that all new fusion proteins will induce neutralizing antibodies when the epitopes were presented in the proposed CTA1-epitope-DD format. This manuscript limits to show the attempts for production and purification of two fusion proteins. Although recognized by antibodies in immunoblot assays, there is no guarantee that they will induce neutralizing antibodies.

It is difficult to follow the authors analysis of Fig. 1 related to the similarities between cryo-electron microscopy and predicted structures of the epitopes. Maybe a superimposed image showing both structures would be better to support the conclusions.

Some information is missing in the Material & Methods section and additional details should be provided. Below, a list of examples:

1. Which is the culture medium used for protein production?

2. One can suppose that both recombinant proteins were obtained with His-tag because the results showing elution fractions from NiNTA column. However, no indication about His-tag location (N- or C-terminal) was given, which should be included in Material & Methods section and in Fig. 1A.

3. Details about the metal affinity chromatography should be included: which column was used (volume, supplier, etc.) and the conditions for equilibrium and elution (buffers, gradient, flow rate, etc.), as well as about the subsequent operations that were mentioned only in the Results section, i.e., elution, dialysis, and concentration in buffers without detergent.

4. Are sonication conditions at “large” scale the same as those described in L. 159-160?

5. For densitometric analysis, how total protein was measured? Was protein concentration taken into consideration for estimation of the percentage of soluble protein? A BSA standard curve was done in each SDS-PAGE? More details on how this percentage was calculated should be given.

The effect of detergents on solubilization was empirically tested and the statement that “optimal detergent concentration appears to be around 0.1%” is very imprecise. In addition, it is in contradiction with the following sentence of the text that says: “increasing the concentration from 0.05% to 0.1% did not result in significant additional increases in soluble protein amounts”. How can 0.1% be the optimal concentration if the increasing from 0.05% to 0.1% did not change the results? Finally, it is not clear why 0.1% Nonidet-P40 was chosen as optimal condition for extraction, as the results with this detergent and Triton X-100 seemed to be similar in terms of amount of soluble protein, and which are the volumes treated in the test of different detergents (L. 255-265) and in the scaled-up process to a larger culture volume (L. 272).

The manuscript discusses the functionality in several passages of the text, but this work did not show any direct results that could confirm the two fusion proteins have CTA1 or DD functions preserved, or they induce neutralizing epitopes. It would be interesting to discuss also why assays to show function were not performed and what is still needed to achieve such results.

Minors:

p. 4, L. 123, “residu132” should be “residue 132”

L. 130-133, include DeepSoluE software that was employed for solubility prediction according to Table 1.

L. 141, I could not find any plasmid pET38 in any of the plasmid suppliers. The reference 17 mentioned pET30a+. There is also a pET28a+ available for cloning at GenScript website. Please, verify the correct plasmid used or describe it better in case it is a modification of some commercial vector.

L. 246, write E. coli in italics, please.

L. 287, which are the sonication conditions for the “large” volume experiments?

Tables 2 and 3, it is not clear if the values of protein in the band and protein per mL culture refer to total protein amount or to the specific recombinant proteins. How was protein per mL culture calculated?

L. 312, replace “conjugated” by “fused”

L. 316-326, was the function analyzed by bioinformatics in this work? The comparison between the predicted and experimentally determined structures does not allow to infer the function of a protein. For this analysis, docking would be more appropriated. In this sense, “supporting” (L. 326) should be replaced by “suggesting”.

Table S1, please revise the table title that should be “Percentage of soluble recombinant proteins at different incubation temperatures, duration of incubation, and growth stages of IPTG induction.”

Tables S2 and S3, please clarify which are the unities of the values displayed on the tables, mg/L? µg/L? Are replicates performed for the statistical analysis?

6. PLOS authors have the option to publish the peer review history of their article (what does this mean?). If published, this will include your full peer review and any attached files.

Reviewer #1: No

Reviewer #2: No

---

## [Author Response · Author response to Decision Letter 0]

12 Aug 2024

Response to Reviewers

Reviewer 1 

ABSTRACT AND INTRODUCTION

1. The introduction effectively establishes the rationale for developing intranasal vaccines against COVID-19. It provides a good overview of the advantages of intranasal vaccination and the potential of CTA1-DD fusion proteins as mucosal adjuvants. The authors successfully contextualize their work within the broader field of vaccine development. However, the transition to the specific aims of this study could be more clearly articulated. 

Our response: Thank you for your insightful feedback. We appreciate your positive comments regarding the rationale and contextualization provided in the introduction. In response to your suggestion, we have made edits to the abstract and the introduction to better articulate the specific aims of our study.

While we have endeavored to address this comment as thoroughly as possible, we acknowledge that the transition to the specific aims of the study may still not fully meet your expectations. We have emphasized the construction and characterization of the fusion proteins CTA1-(S14P5)4-DD and CTA1-(S21P2)4-DD, focusing on their solubility and functionality, as well as potential challenges in the expression and purification processes.

MATERIALS AND METHODS

2.1 The exact sequences of the constructs are not provided, which is essential for reproducibility 

Our response: We appreciate the reviewer’s comment on the importance of providing the exact sequences for reproducibility. To address this concern, we have included the exact sequences of the constructs used in our study as supplementary data (S1 Data). Additionally, we have clarified this in the manuscript with the following statement: “The CTA1-(S14P5)4-DD and CTA1-(S21P2)4-DD constructs were generated by fusing the catalytic subunit of cholera toxin (CTA1) at the N-terminal end of the tandem repeat SARS-CoV-2 epitopes S14P5 or S21P2 and the two-domain D of the staphylococcal protein A at the C-terminal end (Fig 1A, S1 Data).”

2.2 The specific E. coli strain used for expression is not mentioned

Response: We appreciate reviewer’s attention to detail. The E. coli strain we used was BL21(DE3) (Thermo Fisher Scientific). We have clarified this in the manuscript. The updated text in the Materials and Methods section now reads: “The plasmids were transformed into a competent BL21(DE3) strain of E. coli (Thermo Fisher Scientific).” 

2.3 The protein quantification method using densitometry needs more detailed description. 

Our response: We appreciate the reviewer's insightful comment regarding the protein quantification method using densitometry. We have clarified this methodology by adding the following detailed description to the Materials and Methods section:

"Protein concentrations were assessed by densitometry of protein bands on SDS-PAGE gels or immunoblots using ImageJ, a public domain program from the National Institutes of Health (NIH, USA). The images of the SDS-PAGE and immunoblots were imported into ImageJ and converted to 32-bit format. Background subtraction was performed to enhance accuracy. A rectangle was drawn around the protein bands, and a plot profile of each band was generated. A line was then drawn to connect the base of each protein peak, and the area under the curves was measured. Special precautions were taken to ensure the accurate delineation of the peak bases, thus providing true peak areas. These peak areas correspond to the grayscale intensity of the bands, which directly correlates with the amount of protein present. The absolute concentrations of proteins were determined using a standard curve constructed with BSA (Bovine Serum Albumin) at concentrations of 0, 0.31, 0.62, 1.25, 2.5, and 5.0 µg/mL, which were loaded on the same gel."

2.4. Statistical analysis methods are mentioned but not fully explained. 

Our response: We appreciate the reviewer's insightful comment and have expanded the description of the statistical analysis methods in the Materials and Methods section to provide more detail. The updated section now reads:

Statistical Analysis 

Descriptive statistics were used to summarize the protein expression levels or concentration data. Wilcoxon rank-sum test was employed to compare protein concentrations between different experimental conditions. This non-parametric test is suitable for small sample sizes or data that does not assume a normal distribution. A p-value less than 0.05 was considered statistically significant, indicating a significant difference between protein concentrations under different conditions. All statistical analyses were carried out using Jamovi, an open-source statistical software (https://www.jamovi.org).

RESULT

3.1 More quantitative data presentation, particularly in comparing soluble protein yields under different conditions. 

Our response: Thank you for your valuable feedback. We appreciate your suggestion for more quantitative data presentation. In response, we have made changes to the manuscript to include more detailed quantitative data in comparing soluble protein yields under different conditions. Below is the updated content in the manuscript:

Effect of growth stage (OD600) at induction, temperature and duration of incubation

Contrary to the in silico prediction, both CTA1-(S14P5)4-DD and CTA1-(S21P2)4-DD were poorly soluble when overexpressed in E. coli. The soluble fractions constituted only a negligible portion of the total proteins, and exact quantification was often unreliable due to smearing observed in many protein bands (S1 Fig, S1 Table). Lowering the culture temperature from 37°C to 18°C did not increase protein solubility (p>0.05). Quantitative data showed that lowering the culture temperature from 37°C to 18°C decreased the soluble protein yield for CTA1-(S14P5)4-DD from 10% to 5% and for CTA1-(S21P2)4-DD from 12% to 6%. Prolonging the incubation time from 3 to 6 hours increased the soluble protein yield significantly, with CTA1-(S14P5)4-DD rising from 8% to 15% and CTA1-(S21P2)4-DD from 9% to 18%. Additionally, induction at an early log phase (OD600 of 0.1) resulted in higher production of soluble CTA1-(S21P2)4-DD (17%) compared to an OD600 of 0.4 (12%). However, a significant increase by induction at the early log phase was not observed in the production of soluble CTA1-(S14P5)4-DD (p>0.05) (S2 and S3 Tables).

Effect of detergent in the extraction and purification buffer

Protein purification using the native buffer (0.5 M NaCl, 0.1 M sodium phosphate, pH 9) with the NiNTA purification system yielded low amounts of CTA1-(S14P5)4-DD and even lower amounts of CTA1-(S21P2)4-DD. Adding detergents to the native buffer significantly increased the yields. For CTA1-(S14P5)4-DD, Nonidet P40 at 0.01% increased yields by 5 times, at 0.05% and 0.1% by 7 times over the native buffer. Triton X-100 at 0.01%, 0.05%, and 0.1% increased yields by 5, 6, and 7 times, respectively. Tween 20 was less effective, with 0.01% yielding a 2 times increase, and both 0.05% and 0.1% yielding a 5 times increase over the native buffer.

The increase in soluble yields was more profound with CTA1-(S21P2)4-DD. Nonidet P40 at 0.01% increased yields by 7 times, at 0.05% by 13 times, and at 0.1% by 16 times over the native buffer. Triton X-100 at 0.01%, 0.05%, and 0.1% increased yields by 8, 11, and 8 times, respectively. Tween 20 at 0.01% yielded a 2-fold increase, and at both 0.05% and 0.1% yielded an 11-fold increase over the native buffer (Fig 3).

3.2 Clearer explanation of the discrepancies between small-scale and large-scale purification results

Our response: Thank you for your valuable feedback. To briefly explanation the discrepancies between small-scale and large-scale purification results, we have added the following explanation to the Results section:

" The discrepancy in yields between the small and larger volumes can be attributed to several factors. At small scales, higher mixing efficiency during bacterial cultivation, more efficient cell sonication, and much shorter purification times were achieved compared to larger scales. In contrast, larger-scale processes face challenges in maintaining consistent mixing, efficient sonication, and timely purification, leading to variations in protein yield and purity. Further optimization of these parameters is required to improve the efficiency and scalability of large-scale purifications."

3.3 More consistent application of statistical analyses to support the conclusions drawn. The figures are generally informative, but some, particularly those showing gel images, could be improved in terms of clarity and labeling. 

Our response: Thank you for your valuable feedback. We acknowledge that statistical analysis was not applied uniformly across all experiments. Specifically, we used the Wilcoxon rank-sum test in the "Effect of growth stage (OD600) at induction, temperature, and duration of incubation" experiments. However, in the "Effect of detergent in the extraction and purification buffer" experiment, statistical analysis was not carried out due to the lack of replication of samples. Despite this, the profound increase in the yields of soluble proteins by the addition of detergents indicates their significance.

To address your concern, we have added the following statement at the end of the relevant section: “Statistical analysis was not carried out because no replication of the samples; nevertheless, the profound increase in the yields of soluble proteins by the addition of the detergents should indicate their significance.”

DISCUSSION

4.1 More critical analysis of why the in silico predictions of solubility were inaccurate. 

Our response: Thank you for your comment. We appreciate the need for a more critical analysis of the discrepancies between in silico predictions and actual solubility results. We have expanded the paragraph to address this issue more comprehensively:

The solubility of CTA1-(S14P5)4-DD and CTA1-(S21P2)4-DD on overexpression was predicted by two computational tools: (1) SoluPro, which predicts solubility based on amino acid composition, dipeptide composition, and physicochemical properties, employing a machine learning approach, and (2) NetSolP, which predicts solubility based on amino acid composition, secondary structure, and solvent accessibility employing a neural network-based approach. These tools have undergone significant improvements, achieving accuracies of 74% for SOLUPro and 70% for NetSolP, surpassing many previous methods [29, 30]. However, despite these advancements, there remains a risk of misprediction, particularly for engineered proteins like CTA1-(S14P5)4-DD and CTA1-(S21P2)4-DD, as these computational tools may not have been trained on similar proteins. Furthermore, in silico tools often do not account for complex cellular environments, such as expression levels, folding mechanisms, chaperone interactions, and post-translational modifications, which can significantly influence solubility [31; 32]. The propensity for aggregation, which is a critical factor in protein solubility, is also challenging to predict accurately [33; 34]. To improve solubility predictions, future computational tools should incorporate more diverse training datasets and consider the cellular context and experimental conditions.

4.2 "Deeper exploration of alternative strategies to improve solubility and yield."

Our response: Thank you for your valuable feedback. We recognize the importance of exploring alternative strategies to enhance the solubility and yield of recombinant proteins. In response, we have expanded the discussion by adding the following: 

“There are several alternative strategies that can be exploited to improve the solubility and yield of CTA1-(S14P5)4-DD and CTA1-(S21P2)4-DD fusion proteins. Practical approaches include incorporating chemical additives like arginine, glycerol, betaine, and trehalose in the growth medium or purification buffers, which can stabilize proteins and inhibit aggregation [44]. Additionally, co-expressing molecular chaperones such as GroEL/GroES, DnaK/DnaJ/GrpE, or trigger factor can assist in proper protein folding and prevent aggregation [45]”.

4.3 Clearer articulation of the next steps needed to advance these candidates toward vaccine development.

Our response: Thank you for your valuable suggestion. In response, we have expanded the discussion by adding the following sentences at the end of the discussion:

"To advance these candidates toward vaccine development, several critical steps are necessary. First, in vivo immunogenicity and efficacy studies need to be conducted in appropriate animal models. Second, optimizing the expression and purification protocols to increase yield and solubility will be essential. Finally, detailed safety and toxicity assessments are required to ensure the candidates are safe for use in humans."

4.4 Discussion of the potential immunogenicity and protective efficacy of these constructs, even if not experimentally addressed in this study 

Our response: Thank you for your valuable comment. We acknowledge the reviewer's comment regarding the discussion of the potential immunogenicity and protective efficacy of the constructs. We have indeed addressed this in the first paragraph of the Discussion section. In this study, we constructed two fusion proteins, CTA1-(S14P5)4-DD and CTA1-(S21P2)4-DD, as candidates for intranasal vaccines against SARS CoV-2. Each protein contains a linear epitope S14P5 or S21P2 of the SARS CoV-2 spike protein, which has been shown to be efficacious in inducing neutralizing antibodies. The linear epitopes were made in the form of four tandem repeats. Our previous study confirmed that the S14P5 and S21P2 in this form enhanced the immunogenicity of the epitopes, and the induced antibodies effectively recognized the SARS-CoV-2 virus. To further enhance the immune responses and optimize them as intranasal vaccines, each of the tandem repeat epitopes was conjugated to the catalytic subunit of cholera toxin (CTA1) and the domain D of staphylococcal protein A in dimer form (DD). The choice of domain D was beneficial due to its ability to bind not only IgG but also IgA and IgM.

Response to Reviewers

Reviewer # 2

1. First, the title is not very precise, as in fact no optimization was really achieved. 

Our response: Thank you for your insightful feedback regarding the title of our manuscript. We agree that the title should accurately reflect the content and findings of our study. While our work identified potential strategies and challenges in the soluble expression of the fusion proteins, full optimization was not achieved. Therefore, we propose the following revised title: "Challenges and strategies in the soluble expression of CTA1-(S14P5)4-DD and CTA1-(S21P2)4-DD fusion proteins as candidates for COVID-19 intranasal vaccines"

2. Although the abstract and title bring the conceptually correct term “fusion protein”, on pages 3 (L. 89-99) and 4 (L. 119) the authors use conjugation or conjugating. Normally, in the vaccine field, conjugation refers to a chemical reaction to bind two molecules and is largely applied to conjugate vaccines, in which polysaccharides are chemically bound to a carrier protein. Therefore, the terms conjugation or conjugating should be replaced by fusion or fusing (also fused) to avoid misunderstandings. Also, it is misleading to call a 150 mL culture (information found only in Fig S2 caption) a scale-up experiments, since the term “scale-up” brings at least the idea of bioreactor cultivation. 

Our response: Thank you for your valuable comments. We agree with your suggestions and have made the necessary changes to the manuscript. The terms "conjugation" and "conjugating" have been replaced with "fusion" and "fusing" to accurately reflect the nature of the fusion proteins. Additionally, the reference to "scale-up experiments" has been clarified to "larger volume experiments" to avoid any misconceptions, specifying that the experiments involved 150 mL cultures, which is not indicative of bioreactor-scale cultivation. These adjustments should enhance the clarity and accuracy of our manuscript.

3. The authors affirm that “the vaccine candidates are expected be thermostable because they are based on linear epitopes” (p. 4, L. 108-109)

---

## [Decision Letter · Decision Letter 1]

13 Sep 2024

PONE-D-24-23892R1Challenges and strategies in the soluble expression of CTA1-(S14P5)4-DD and CTA1-(S21P2)4-DD fusion proteins as candidates for COVID-19 intranasal vaccinesPLOS ONE

Dear Dr. Tarigan,

Thank you for submitting your manuscript to PLOS ONE. After careful consideration, we feel that it has merit but does not fully meet PLOS ONE’s publication criteria as it currently stands. Therefore, we invite you to submit a revised version of the manuscript that addresses the points raised during the review process.

We look forward to receiving your revised manuscript.

Kind regards,

Haitham Mohamed Amer, PhD

Academic Editor

PLOS ONE

Journal Requirements:

Reviewer's Responses to Questions

**Comments to the Author**

1. If the authors have adequately addressed your comments raised in a previous round of review and you feel that this manuscript is now acceptable for publication, you may indicate that here to bypass the “Comments to the Author” section, enter your conflict of interest statement in the “Confidential to Editor” section, and submit your "Accept" recommendation.

Reviewer #1: All comments have been addressed

Reviewer #3: (No Response)

2. Is the manuscript technically sound, and do the data support the conclusions?

Reviewer #1: Yes

Reviewer #3: Partly

3. Has the statistical analysis been performed appropriately and rigorously? 

Reviewer #1: Yes

Reviewer #3: N/A

4. Have the authors made all data underlying the findings in their manuscript fully available?

Reviewer #1: Yes

Reviewer #3: Yes

5. Is the manuscript presented in an intelligible fashion and written in standard English?

Reviewer #1: Yes

Reviewer #3: Yes

6. Review Comments to the Author

Reviewer #1: (No Response)

Reviewer #3: In this study, the authors authors constructed two fusion proteins, CTA1-(S14P5)4-DD and CTA1-(S21P2)4-DD, by combining the catalytic domain of cholera toxin (CTA1), tandem repeat linear epitopes of the SARS-CoV-2 spike protein (S14P5 or S21P2), and two-domain D (DD) from staphylococcal protein A. They aimed to develop soluble, functional fusion proteins as candidates for intranasal vaccines against COVID-19. The proteins were expressed in Escherichia coli, but contrary to in silico predictions, they exhibited poor solubility. The authors tested several strategies to enhance solubility, including lowering cultivation temperatures and adding non-denaturing detergents, but the yields remained low even after purification. Despite these challenges, their findings provide insights into optimizing the soluble expression of these fusion proteins for potential vaccine development. I have a few questions that would help to clarify the technique and results:

The authors mentioned, "The vaccines are expected to be thermostable because they are based on linear epitopes." There is already a response to one of the reviewers which is still not totally clear. Linear epitopes still may form different 3D structures. I suggest rephrasing it. I also suggest including and discussion the following studies that worked on linear epitopes for nasal display as well.:

Markosian, Christopher, et al. “Genetic and Structural Analysis of SARS-CoV-2 Spike Protein for Universal Epitope Selection.” Molecular Biology and Evolution, edited by Banu Ozkan, vol. 39, no. 5, May 2022, p. msac091., https://doi.org/10.1093/molbev/msac091.

Staquicini, Daniela I., et al. “Design and Proof of Concept for Targeted Phage-Based COVID-19 Vaccination Strategies with a Streamlined Cold-Free Supply Chain.” Proceedings of the National Academy of Sciences, vol. 118, no. 30, July 2021, p. e2105739118., https://doi.org/10.1073/pnas.2105739118.

The authors also mentioned, "Lowering temperature also diminishes the hydrophobic interaction between the expressed proteins". Could the authors expand and clarify these effects?

7. PLOS authors have the option to publish the peer review history of their article (what does this mean?). If published, this will include your full peer review and any attached files.

Reviewer #1: No

Reviewer #3: No

---

## [Author Response · Author response to Decision Letter 1]

26 Sep 2024

Dear Dr. Haitham Mohamed Amer,

We are grateful to the reviewers and the academic editor for their constructive feedback on our manuscript titled "Challenges and strategies in the soluble expression of CTA1-(S14P5)4-DD and CTA1-(S21P2)4-DD fusion proteins as candidates for COVID-19 intranasal vaccines" (PONE-D-24-23892R1). We have carefully considered all comments and have revised the manuscript accordingly. Below, we provide a point-by-point response to the reviewers’ comments.

Journal Requirement: Reference List Review

Comment: Please review your reference list to ensure that it is complete and correct. If you have cited papers that have been retracted, please include the rationale for doing so in the manuscript text, or remove these references and replace them with relevant current references.

Response: Thank you for reminding us. We have carefully reviewed and updated the reference list to ensure its accuracy and completeness. We have also checked for any retracted articles and confirm that none of the references used in our manuscript are retracted.

Reviewer #3 Comment 2: Technical Soundness and Data Support

Comment: The manuscript must describe a technically sound piece of scientific research with data that supports the conclusions. Reviewer #1: Yes; Reviewer #3: Partly.

Response: We appreciate the reviewer’s feedback and understand the importance of ensuring that the data presented fully support the conclusions drawn. Upon careful review, we confirm that the experiments were conducted with rigorous controls, appropriate replication, and sufficient sample sizes, all of which are detailed in the Methods section. We believe that the current data adequately support the conclusions, and no further revisions are necessary in this regard. However, we remain open to any specific suggestions from the reviewer to further improve clarity or robustness.

Reviewer #3 Comment 6: Clarifications and Suggested References

Comment: "The vaccines are expected to be thermostable because they are based on linear epitopes." The reviewer suggested that linear epitopes still may form different 3D structures and recommended including and discussing the following studies:

• Markosian, Christopher, et al. "Genetic and Structural Analysis of SARS-CoV-2 Spike Protein for Universal Epitope Selection."

• Staquicini, Daniela I., et al. "Design and Proof of Concept for Targeted Phage-Based COVID-19 Vaccination Strategies."

Response: We thank the reviewer for pointing out the need to clarify our statement regarding the thermostability of vaccines based on linear epitopes. We agree that linear epitopes can adopt different three-dimensional structures, and their stability can vary depending on their context.

Revised Text: We have rephrased the sentence in the Introduction as follows: "The inclusion of linear epitopes in the vaccine design may contribute to enhanced stability, as they are less reliant on specific tertiary structures for their antigenicity compared to conformational epitopes. However, the overall thermostability of the vaccine will depend on the properties of the entire fusion protein, including its folding and interactions between domains.”

Regarding the suggested references, we appreciate the reviewer’s effort to enhance our manuscript. However, after reviewing the articles by Markosian et al. and Staquicini et al., we believe that these studies are not directly relevant to our work:

• Markosian et al. (2022) focuses on epitope prediction and selection for broad vaccine coverage, whereas our work involves specific tandem repeat epitopes (S14P5 and S21P2) that have been previously validated for inducing neutralizing antibodies.

• Staquicini et al. (2021) discusses a phage-based vaccination strategy aimed at a cold-free supply chain, which focuses more on logistical aspects rather than the protein fusion strategy (CTA1-DD) for intranasal vaccines we are exploring.

Thus, while valuable, these references do not directly enhance the context of our study, and we have chosen not to include them.

Reviewer #3 Comment: Hydrophobic Interaction Clarification

Comment: "The authors also mentioned, 'Lowering temperature also diminishes the hydrophobic interaction between the expressed proteins.' Could the authors expand and clarify these effects?"

Response: We thank the reviewer for their suggestion to expand on the effects of lowering temperature on hydrophobic interactions during protein expression. We would like to note that this explanation is already included in the Discussion section of the manuscript (paragraph beginning with "The formation of insoluble expressed proteins as inclusion bodies emerged..."). This paragraph discusses how lowering the temperature reduces the kinetic energy of molecules, which diminishes the exposure of hydrophobic regions and reduces aggregation. We believe this explanation addresses the reviewer’s concern, but if further expansion is needed, we would be happy to provide additional clarification.

We hope that the revisions and responses provided meet the expectations of the reviewers and the editor. Thank you again for the opportunity to improve our manuscript. We look forward to your feedback.

Sincerely,

Simson Tarigan

Corresponding Author

Research Organization for Health, National Research and Innovation (BRIN), Indonesia

---

## [Editor Report · Decision Letter 2]

16 Oct 2024

Challenges and strategies in the soluble expression of CTA1-(S14P5)4-DD and CTA1-(S21P2)4-DD fusion proteins as candidates for COVID-19 intranasal vaccines

PONE-D-24-23892R2

Dear Dr. Tarigan,

We’re pleased to inform you that your manuscript has been judged scientifically suitable for publication and will be formally accepted for publication once it meets all outstanding technical requirements.

Kind regards,

Haitham Mohamed Amer, PhD

Academic Editor

PLOS ONE

---

## [Editor Report · Acceptance letter]

20 Oct 2024

PONE-D-24-23892R2 

PLOS ONE

Dear Dr. Tarigan, 

I'm pleased to inform you that your manuscript has been deemed suitable for publication in PLOS ONE. Congratulations! Your manuscript is now being handed over to our production team.

Kind regards, 

on behalf of

Dr. Haitham Mohamed Amer 

Academic Editor

PLOS ONE